# Stress promotes RNA G-quadruplex folding in human cells

Prakash Kharel [1,5], Marta Fay[1,5], Ekaterina V. Manasova[2], Paul J. Anderson[1,3], Alexander V. Kurkin[2], Junjie U. Guo [4] ✉ & Pavel Ivanov [1,3] ✉

Guanine (G)-rich nucleic acids can fold into G-quadruplex (G4) structures under permissive conditions. Although many RNAs contain sequences that fold into RNA G4s (rG4s) in vitro, their folding and functions in vivo are not well understood. In this report, we showed that the folding of putative rG4s in human cells into rG4 structures is dynamically regulated under stress. By using high-throughput dimethylsulfate (DMS) probing, we identified hundreds of endogenous stress-induced rG4s, and validated them by using an rG4 pull-down approach. Our results demonstrate that stress-induced rG4s are enriched in mRNA 3′-untranslated regions and enhance mRNA stability. Furthermore, stress-induced rG4 folding is readily reversible upon stress removal. In summary, our study revealed the dynamic regulation of rG4 folding in human cells and suggested that widespread rG4 motifs may have a global regulatory impact on mRNA stability and cellular stress response.

G-quadruplexes (G4s) are nucleic acid secondary structures formed by the stacking of two or more G-quartets, each formed by a planar arrangement of four guanines (Gs) connected via H-bonding[1] and further stabilized by monovalent cations such as $K^+$ and $Na^+$ in the central cavity[2] (Fig. 1a). In eukaryotes, potential G4-forming sequences are enriched in the regulatory regions of both genome and transcriptome[3,4] and have been implicated in a variety of cellular processes and human diseases[5,6]. Computational analyses and high-throughput experimental studies of the human transcriptome suggest that thousands of putative rG4s are present within mRNAs[7–9]. These putative rG4 sequences are more prevalent in the 5' and 3' untranslated regions (UTRs) of mRNAs while strongly depleted in the coding sequences (CDS), suggesting potential roles in the regulation of mRNA maturation, transport, and translation[5,6,10]. Indeed, rG4s in 5' UTRs have been shown to modulate translation[11–13], while rG4 sequences in 3' UTRs influence microRNA targeting[14], RNA localization[15], and alternative polyadenylation[12]. Furthermore, recent studies suggest that RNA secondary structures including rG4s can act as seeds to assemble RNA binding proteins (RBPs) and eventually generate ribonucleoprotein (RNP) granules or bodies[16,17]. Collectively,

these findings suggest that rG4 sequences can have a crucial and widespread impact on posttranscriptional gene regulation and RNA metabolism in eukaryotes.

While the accumulating evidence supports a regulatory role for rG4 sequences, whether they act as single-stranded RNA motifs or as folded rG4 structures has been intensely debated. Multiple studies have attempted to detect rG4 folding in cells using different approaches, which have led to apparently distinct conclusions. For example, G4-specific antibodies[18], G4-specific ligands[19–21], and rG4-specific chemical probes[9,22–24] have detected positive signals for the presence of rG4s in human cells[18,25] and in plants[26]. Other methods based on small-molecule probes capable of live cell G4 imaging bring a suitable complementarity to the available rG4 tools[23,27]. On the other hand, studies based on RNA structure probing can provide a more quantitative measurement of rG4 folding in living cells. One such method is based upon the ability of 2-methylnicotinic acid imidazolide (NAI) to selectively acylate the 2′-hydroxyl of 3′-terminal G residues within G-tracts when an rG4 is in a folded state. The resulting 2′-O-acylated nucleotides can cause stalls during reverse transcription (RT) and be quantified by high-throughput sequencing of truncated cDNAs[26,28]. A

[1]Division of Rheumatology, Inflammation, and Immunity, Department of Medicine, Brigham and Women's Hospital, Harvard Medical School, Boston, MA 02115, USA. [2]Chemistry Department of Lomonosov Moscow State University, 119991 Moscow, Russia. [3]Harvard Medical School Initiative for RNA Medicine, Boston, MA 02115, USA. [4]Department of Neuroscience, Yale School of Medicine, New Haven, CT 06520, USA. [5]These authors contributed equally: Prakash Kharel, Marta Fay. ✉e-mail: junjie.guo@yale.edu; pivanov@rics.bwh.harvard.edu

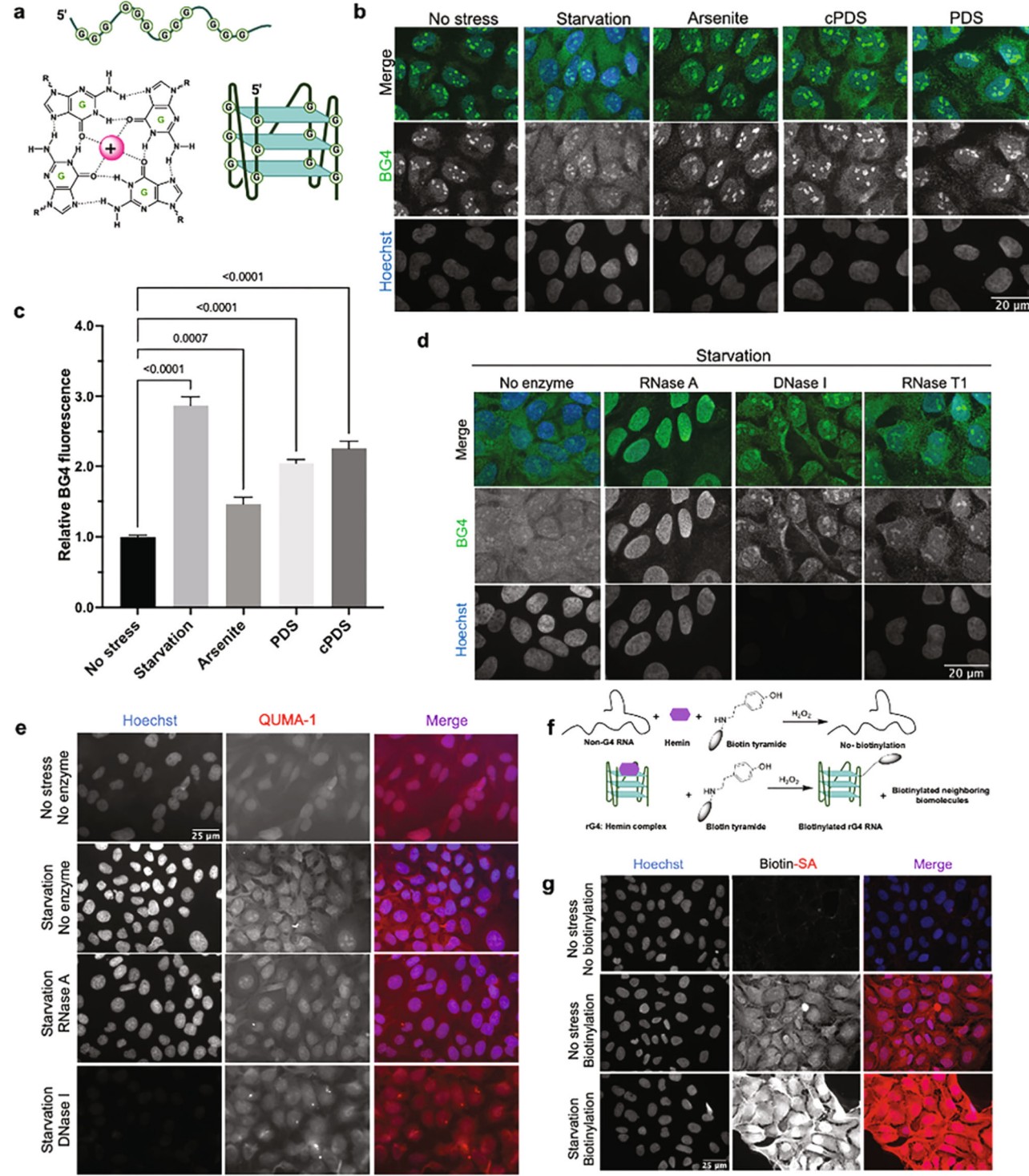

**Fig. 1 | RNA G-quadruplex formation is enhanced under stress in U2OS cells. a** A square planar G-quartet arrangement stabilized by a monovalent cation (H-bonds and cation-oxygen coordination is shown with dotted bonds), and a schematic of 3-tier G-quadruplex structure stabilized by monovalent cations. The schematic depicts 3 G-quartets (light green planes) stacking with RNA-backbone in a parallel orientation. **b** BG4 staining in U2OS cells demonstrated both RNA and DNA can form G4s in U2OS cells, and rG4 folding (cytosolic BG4 fluorescence) in the U2OS cells is increased under various stress conditions as indicated by more intense fluorescence signals of BG4 antibody under 100 μM sodium arsenite mediated oxidative stress and starvation stress, rG4, and G4 stabilizing ligands carboxypyridostatin (cPDS) and pyridostatin (PDS) were used as positive controls. **c** Quantification of fluorescence intensity of BG4 signals, the bars represent the mean +/- SD of fluorescence intensity from 50 cells for each treatment (biological

triplicate), statistical significance was tested by One-way ANOVA using no-stress cells as controls, Multiple comparisons, Dunnet correction, and *p*-values provided. **d** RNase A treatment almost completely abolished the cytoplasmic BG4 signaling, while DNase I treatment minimized nuclear BG4 staining, and RNase T1 treatment had little effect on resultant cytosolic BG4 fluorescence suggesting cellular RNA is folded in rG4 under stress. **e** rG4 specific QUMA-1 fluorescence dye was used as a complementary approach to BG4 staining, fixed cells were treated with 2 μM QUMA-1 for 15 min, and imaged. **f, g** Hemin-bound rG4s have peroxidase activity which can be tuned to biotinylate the corresponding RNA and potentially its interactome. **f** Schematic of rG4-hemin-biotin tyramide reaction, and **g** enhanced biotinylation of starved cells in comparison to the non-stressed cells clearly indicates a higher rG4 folding under stress.

second method uses dimethylsulfate (DMS) to methylate the N7 position of G residues, which is on the Hoogsteen face and only becomes inaccessible when it is involved in H-bonding as part of an rG4 structure[28] (Fig. 1a). Once methylated at N7 position, these G nucleotides can no longer participate in rG4s in vitro, thereby alleviating RT stalling by folded rG4 structures. Both methods have previously been applied to the poly(A)$^+$ transcriptome. The results have shown that, although a large fraction of mammalian mRNAs contain one or more regions that can readily assemble rG4s in vitro, they are overwhelmingly unfolded in yeast and mammalian cells[28]. Because DMS and SHAPE reagents at high concentrations can also potentially influence G4 folding in cells[29], the procedure must be calibrated such that the measurement stays within the dynamic range. Recent studies using the same chemical probing methods have shown substantial levels of folded rG4s in *Arabidopsis*[26,30], further validating that chemical probing does not preclude the detection of rG4s in vivo. Collectively, previous studies have led to a revised model of rG4 folding in mammalian cells in which a large number of potential rG4 sequences may undergo transient folding and unfolding.

The prevalence of mammalian rG4s and their predominantly unfolded steady state under normal conditions raise the possibility that their folding may be regulated by cellular and/or environmental stress signals. In this work, we found that rG4 folding was induced by stress in human cells. Using DMS probing followed by high-throughput sequencing and RT stall analysis, we identified hundreds of folded rG4s in cells under stress. We developed and applied a small-molecule probe to selectively pull down rG4s from the cellular transcriptome and validated our findings of stress-induced rG4s. To elucidate the functions of stress-induced rG4 folding, we found that mRNAs containing 3′UTR rG4 structures were more stable under starvation. Lastly, stress-induced global rG4 folding was reversed upon the removal of stress. These results suggested dynamic and potentially widespread functions of rG4s in regulating mRNA metabolism as part of the cellular stress response.

## Results

### Elevated rG4 folding in human cells under stress

To test whether rG4 folding is regulated by cellular stresses, human osteosarcoma (U2OS) cells were fixed and probed with a G4-specific antibody (BG4)[25], which showed predominantly nuclear and a relatively low level of cytoplasmic staining (Fig. 1b). When U2OS cells were cultured in starvation media (Hank's balanced salt solution, HBSS) for 2 h before fixation, BG4 signal intensity was significantly increased in the cytoplasm but not in the nucleus (Fig. 1b, c), suggesting that cytoplasmic rG4s became more folded upon starvation. To test whether other cellular stresses could also induce rG4 folding, we induced oxidative stress in U2OS cells by treating them with 100 μM sodium arsenite for 1 h and observed ~2-fold increase in BG4 signal intensity in the cytoplasm (Fig. 1b-c). We also observed a ~2-fold increase in BG4 signal under cold stress (Supplementary Fig. 1). As validations for the specificity of BG4, known rG4-stabilizing ligands carboxypyridostatin (cPDS) and pyridostatin (PDS) also enhanced BG4 staining (Fig. 1b-c). While DNase I treatment reduced BG4 signal in the nucleus, cytoplasmic BG4 signal was resistant to DNase I but was instead sensitive to RNase A treatment (Fig. 1d). In contrast, RNase T1, which targets guanines in unfolded RNA regions, had a minimal effect in reducing BG4 signals, further indicating that the increased cytoplasmic BG4 signal under stress was due to rG4 folding. Similar results were observed using transformed African green monkey kidney COS7 cells (Supplementary Fig. 2).

Since BG4 has been reported to also cross-react to a lesser extent with non-G4 motifs[31], we applied two additional complementary methods to assess whether the observed BG4 signal indeed represented folded rG4s in cells. First, we stained cells with a G4-specific QUMA-1 dye[23], which showed enhanced signals in cytoplasm upon starvation (Fig. 1e). As expected, cytoplasmic QUMA-1 signal was sensitive to RNase A but resistant to DNase I treatment (Fig. 1e). Second, we utilized the inherent peroxidase-like activity of hemin-bound G4s[32,33] to further confirm the stress-responsive nature of rG4s in human cells (Fig. 1f). When sensitized with a peroxide, rG4-hemin complexes can cause self-biotinylation. After validating the G4-specific biotinylation activity of RNA-hemin complexes in vitro (Supplementary Fig. 3), we treated U2OS cells with hemin, biotin-tyramide, and H$_2$O$_2$, and quantified the level of biotinylation by using a streptavidin (SA) antibody (Fig. 1g). Indeed, increased biotinylation signals were observed in starved cells, confirming our finding of stress-induced rG4 folding.

### Identification of stress-induced rG4s by DMS probing

To systematically investigate rG4s that are folded in U2OS cells under stress, we first used an RT-stop profiling approach to identify endogenous rG4 regions in the transcriptome[28] (Fig. 2a). During primer extension, rG4s cause RT stalling in a K$^+$-dependent manner, generating truncated cDNA fragments that can be sequenced in a high-throughput manner. As expected, strong RT stops (≥10-fold higher than average RT stop counts within the same transcript) were highly enriched (70%) at G nucleotides (Fig. 2b), indicating that rG4s present the primary barrier for RT. Consistent with the stabilization of rG4 structures by K$^+$, the fraction of RT stops at G nucleotides was even higher (86%) among the 2,224 RT stops that were ≥2-fold stronger in K$^+$ compared to Na$^+$. In agreement with a previous study[8], rG4 regions were significantly enriched in 3′ UTRs, with 72% rG4 RT stops compared to 42% of all RT stops residing in 3′ UTRs (Fig. 2c).

Next, we quantified the folding states of these endogenous rG4 regions in vitro and in vivo using DMS probing. As mentioned above, DMS methylates the N7 positions of G nucleotides that are accessible and not engaged in G-quartets, thereby reducing RT stalling during primer extension. In contrast, the N7 positions in G-quartets are protected from DMS (Fig. 2a). Indeed, polyA-selected mRNAs folded and DMS-modified in the presence of 150 mM K$^+$ caused on average 1.6-fold stronger RT stops than those folded and DMS-modified in the presence of 150 mM Li$^+$ (Fig. 2d). 335 out of 1,346 (25%) rG4 regions showed ≥2 fold difference in DMS accessibility between K$^+$ and Li$^+$ conditions (Fig. 2d, black), which we carried forward for in vivo DMS probing analysis.

To quantify rG4 folding in vivo, we performed DMS probing in U2OS cells under either basal or starvation conditions and calculated the in vivo folding scores for each of 196 rG4 regions with sequencing coverage under both conditions. Consistent with previous results[28], the vast majority of rG4 regions in unstressed cells in vivo were highly accessible to DMS modifications, with a median folding score of −0.01, indicating that they were in a predominantly unfolded steady state similar to the in vitro state without K$^+$. Strikingly, we observed strong protection of rG4 regions from DMS modifications after starvation, resulting in a median in vivo folding score of 0.64, suggesting that many rG4s became substantially more folded (Fig. 2e). Starvation-induced rG4 folding was widespread, with 72% rG4s showing an increase of in vivo folding scores of ≥0.25 (Fig. 2f). mRNAs containing starvation-induced folded rG4s were enriched in those encoding RNA-binding proteins, ubiquitin ligases, and both cytosolic and membrane proteins (Fig. 2g). Collectively, our unbiased DMS-seq analysis showed that starvation-induced widespread and stable folding of rG4 structures, raising the possibility that they may regulate mRNA metabolism in a stress-dependent manner.

### In vitro validation of rG4 formation by selected mRNA oligos

To verify the capability of DMS-seq-identified mRNA 3′ UTR G-rich regions to fold into rG4s, we analyzed the circular dichroism (CD) spectroscopy characteristics of respective RNA oligos derived from identified putative rG4s under rG4-permissive and non-permissive conditions. All tested candidates showed expected CD characteristics

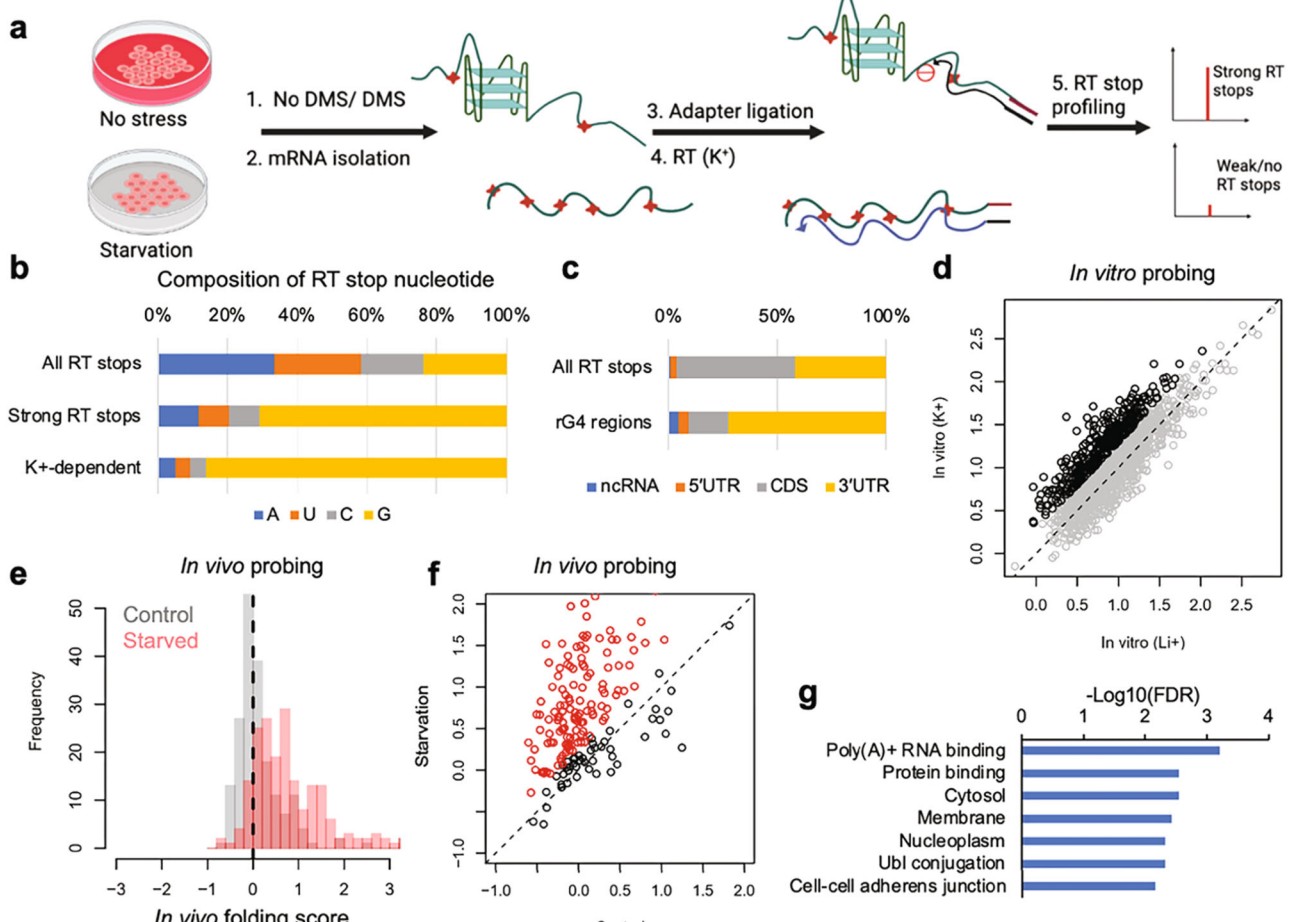

**Fig. 2 | DMS-mediated unbiased structure probing shows mRNA rG4s are folded under stress in vivo. a** A schematic of DMS-seq and RT-stop profiling to identify the bonafide rG4s in U2OS cells under starvation stress. **b** RT stops are highly enriched at G residues. **c** RT stops are selectively enriched in mRNA 3'UTRs. **d** On average, RT stops in 150 mM K+ folding followed by DMS-modified mRNA are 1.6-fold stronger than that in 150 mM Li+ folding followed by DMS-modified mRNA. rG4 regions showing ≥2 fold difference in DMS accessibility between K+ and Li+ conditions are shown in black. **e, f** in vivo rG4 folding increases significantly in starved U2OS cells as compared to unstressed cells. **g** mRNAs containing starvation-induced folded rG4s are enriched in those encoding RNA-binding proteins, ubiquitin ligases, and both cytosolic and membrane proteins.

of rG4s under rG4-permissive conditions (Supplementary Fig. 4). In addition, we performed electromobility gel shift assays and verified their G4 forming ability by using a G4-specific dye, N-methyl mesoporphyrin IX (NMM), which selectively binds parallel G4s and can detect rG4s in acrylamide gels[34,35]. Surprisingly, NMM stained some rG4-oligo bands under both G4-permissive (K+) and non-permissive (Li+) conditions. In fact, these oligos were stained by NMM even under denaturing conditions, suggesting the metastable nature of these rG4s (Supplementary Fig. 5 and Fig. 3b). Additionally, we performed NMM-based fluorescence assays, which showed enhanced fluorescence of rG4-NMM complexes (Supplementary Fig. 6), further supporting the rG4-forming ability of candidate rG4s.

### Biotinylated N-methyl mesoporphyrin IX (bNMM) as a tool to capture cellular rG4s

To validate DMS-seq-identified folded rG4 candidates, we developed a probe that can capture folded rG4s from cell lysates. Because NMM preferentially binds to parallel G4s, we reasoned that biotin modification of NMM could provide a convenient tool for selective pull down of folded rG4 from total RNAs. A similar approach has been used to selectively pull down parallel DNA G4s[36]. Two biotinylated derivatives of NMM (bNMM), namely a single-biotinylated NMM (b[1]NMM) and a double-biotinylated NMM (b[2]NMM) were synthesized (Fig. 3a) with 18-atom long linker(s) to provide enough space to interact with

large RNA molecules (Synthesis and characterization details are provided in supplementary materials).

To test whether the biotinylated derivatives impacted the ability of NMM to recognize the rG4s, we first compared their fluorescence behavior when bound to 5' tiRNA[ala] (tRNA[ala]-derived stress-induced 5' RNA) rG4s, which have previously been used as a benchmark to study rG4-NMM interactions[37]. b[2]NMM showed comparable fluorescence behavior with that of unmodified NMM (Fig. 3c), as demonstrated by a comparable selectivity in rG4 recognition for gel staining (Fig. 3b) and was chosen for further downstream applications. To test the utility of bNMM as rG4 pull-down probes, we compared the binding and elution behavior of previously well-characterized rG4s–(5' tiRNA[ala] rG4[35] and MT3 matrix metalloproteinase mRNA rG4– MMP[38]), and control oligos (Supplementary Table 1). The oligos were folded under G4 permissive (K+) or non-G4 permissive (Li+) conditions and allowed to equilibrate with streptavidin-agarose bound bNMM (bNMM beads). After several washes, the bound RNA was eluted and spectrophotometrically quantified. As presented in Fig. 3d, we observed a significant enrichment of bNMM pulled-down oligos under G4-permissive conditions, validating the selectivity of streptavidin-bound bNMM beads to pull down rG4s. Next, we used this approach to capture rG4-containing mRNAs from stressed and unstressed cells with or without DMS treatment. The spectrophotometric quantification of bNMM-bound and eluted mRNAs showed higher amount of RNA recovered from

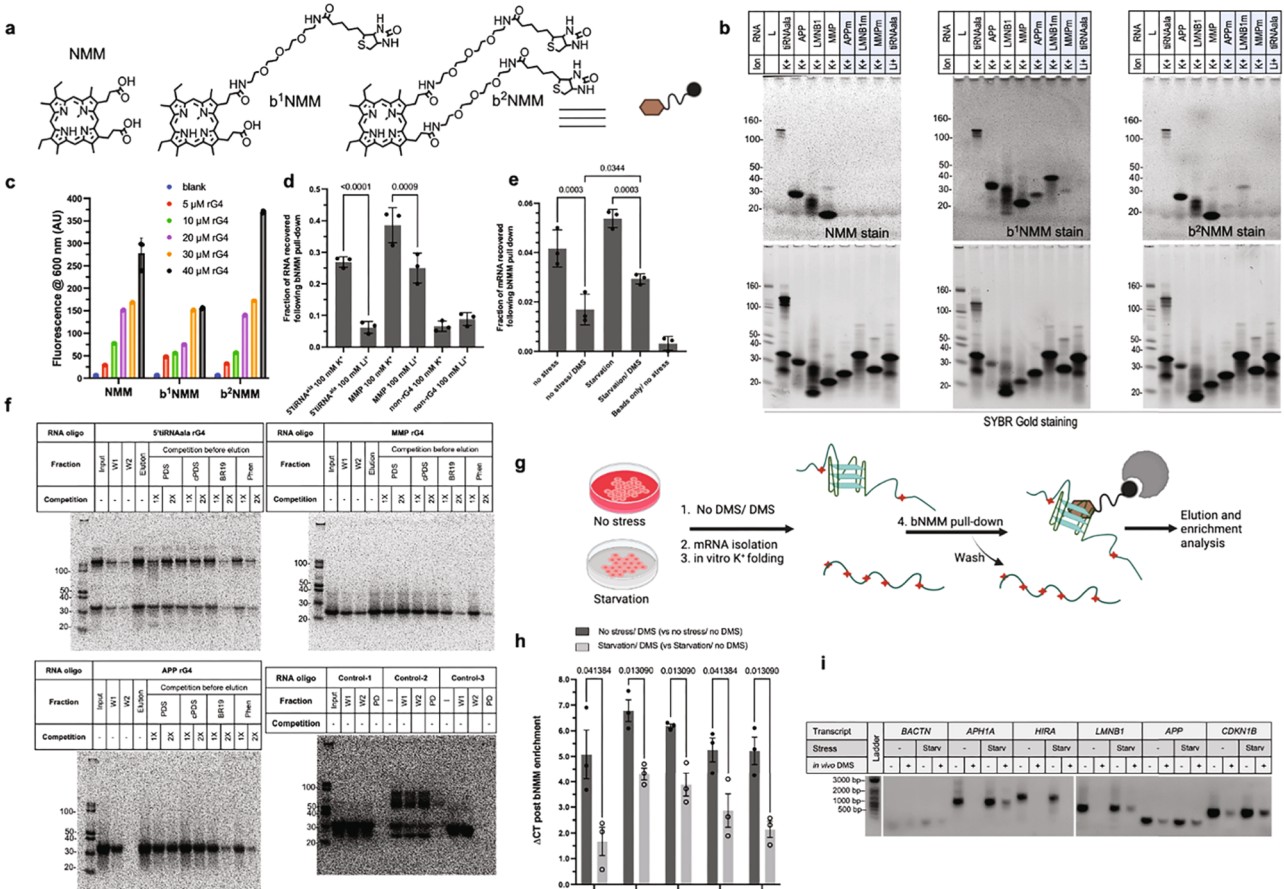

**Fig. 3 | Biotin-functionalized NMM (bNMM) as a selective probe to pull down RNA G-quadruplexes. a** Chemical structure of NMM, a parallel G4 specific porphyrin (right) and biotin functionalized NMMs (synthesis and characterization in SI). **b** gel staining demonstrates double biotinylated NMM shows similar rG4 staining behavior as its parent compound while the single biotinylated version is less selective, ladder represents RNA size marker with molecular weight in Dalton. **c** Biotin functionalized NMMs show similar G4 affinity as their parent compound in NMM fluorescence assay, 5' tiRNA[ala] G4 is used for the selectivity test since 5' tiRNA[ala]-NMM interaction characterization has been already established, data are presented as mean values +/- SD, $n = 3$. **d** Validation of bNMM-based rG4 pull-down approach using synthetic rG4 and control oligos, data are presented as mean values +/− SD, $n = 3$, ordinary One-way ANOVA, pairwise comparison, and adjusted $p$-values are provided. **e** Validation of bNMM-rG4 pull-down approach using cellular mRNA. This also affirms that more Gs are protected from DMS reactivity under starvation stress, data are presented as mean values +/− SD, $n = 3$, ordinary One-way ANOVA, pairwise comparisons, and adjusted $p$-values are provided. **f** Competition of G4 bound bNMM with several G4 interacting small molecules.

BRACO19 and PhenDC3 were able to compete off bNMM from the bNMM-rG4 complex in dose-dependent manner (1X = 2.5 μM, 2X = 5 μM). On the other hand, PDS and cPDS were unable to compete off bNMM under the test conditions (W1-wash 1, W2- wash 2, BR19- BRACO-19, Phen- PhenDC3), ladder represents RNA size marker with molecular weight in Dalton. **g** Schematic of cellular rG4 enrichment and their quantification. Cellular mRNA (with or without the starvation stress and with (+) and without (-) the DMS reaction) was isolated and folded in presence of K⁺ before mixing with bNMM beads. The captured mRNA pool was eluted, and reverse transcribed, and the enrichment was quantified using qPCR and PCR. **h** Candidate mRNAs from DMSeq experiments show a higher abundance in the mRNA isolated from DMS-treated stressed cells, clearly indicating their folding into rG4s in vivo. A smaller difference in ΔCT values under starvation condition indicated rG4 enrichment under such conditions. Data are presented as mean values +/− SD, $n = 3$, Multiple unpaired $t$-tests, and q values are provided. **i** RT-PCR amplicons were run in the agarose gel electrophoresis to visualize the difference in the bNMM bound mRNA under similar conditions as in **h**, ladder represents DNA base pair molecular marker with weight in Dalton, source image Supplementary Fig. 7.

---

stressed cells, consistent with the finding that cellular rG4s were more folded under starvation conditions (Fig. 3e). To further validate that the interaction of bNMM with the G-rich RNA was truly based on NMM-rG4 interaction, we performed competition experiments where bNMM-rG4 complexes were challenged with known rG4-interacting ligands before the elution step (Fig. 3f). While PDS and cPDS were unable to efficiently compete rG4s off of bNMM (presumably due to differences between their binding affinities towards tested rG4s, and/or due to the difference in the binding sites), BRACO-19 and PhenDC3 ligands competed off NMM from rG4-NMM complexes (Fig. 3f).

If an rG4 region is folded in vivo, the N7 position of constituent Gs would be protected from DMS, which in turn allows such sequence to fold back to rG4 under G4 permissive condition (150 mM K⁺) in vitro. Consequently, bNMM capture of rG4-containing RNAs from total mRNA isolated from unstressed cells with or without the DMS treatment versus those from the stressed cells, and their subsequent

elution and quantification provides a snapshot of relative rG4 folding of individual mRNA in vivo. As schematically represented in Fig. 3g, total mRNA was isolated from U2OS cells under various conditions and was folded in presence of 150 mM KCl. Folded total mRNA was incubated with bNMM beads to capture rG4-bearing mRNAs. Bound mRNA was eluted, and the relative abundance of bNMM-bound mRNA was determined using reverse transcription-quantitative polymerase chain reaction (RT-qPCR) and RT-PCR (Fig. 3h-i). Consistent with our DMS-seq results, these experiments showed that a higher fraction of rG4-forming mRNAs were inaccessible to DMS during starvation and subsequently captured by bNMM.

## mRNA stabilization by 3′UTR rG4s under stress

Because 3′ UTR structural elements are known to regulate mRNA decay[39,40], we hypothesized that the induced folding of 3′ UTR rG4s might regulate the stability of those transcripts under stress. To test

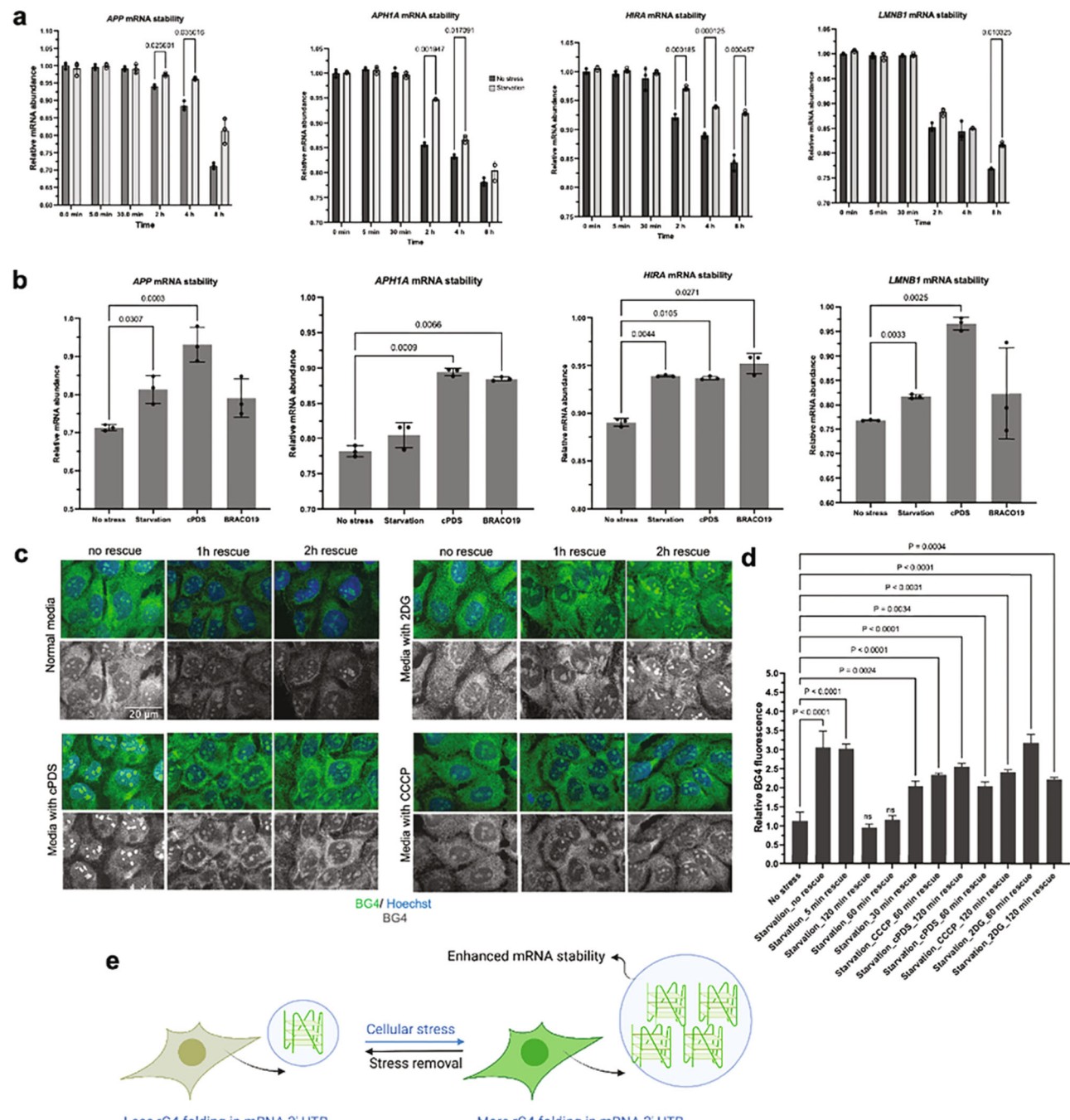

**Fig. 4 | 3′ UTR RNA G-quadruplex folding enhances mRNA stability under stress and such folding is reversible. a** U2OS cells were actinomycin-D treated over an indicated period of time and the relative abundance of specific mRNA was analyzed. RT-qPCR quantification showed 3'UTR rG4 bearing mRNAs are significantly more stable under starvation stress, data are presented as mean values +/− SD, $n = 3$, Multiple paired $t$-tests, $p$-values are provided when $p > 0.05$. **b** Comparative mRNA stability in the presence of rG4 stabilizing ligands (cPDS and BRACO-19) showed a similar trend in mRNA stability as observed under starvation conditions, data are presented as mean values +/− SD, $n = 3$, ordinary One-way ANOVA, Multiple comparisons, adjusted $p$-values are provided when $p > 0.05$. **c** rG4 folding is reversible

upon stress removal (also Supplementary Fig. S9) and such reversibility can be largely paused by rG4 stabilizing ligand cPDS, along with nutrient stresses created by 2-DG (glycolysis inhibitor) and CCCP (oxidative phosphorylation inhibitor) treatment. **d** Quantification of BG4 fluorescence from 10 cells per conditions in **c** and Supplementary Fig. S9, the experiment was performed in biological triplicate, the data presented as mean values +/− SD. Ordinary One-way ANOVA, Multiple comparisons, $p$-values are provided when $p > 0.05$, and **e** proposed model to demonstrate the dynamic nature of stress-responsive rG4 folding and their contribution to mRNA stability.

our hypothesis, we treated the cells with actinomycin D (ActD) (an RNA polymerase II inhibitor) for different durations and collected RNA over those time intervals. We observed that 3' UTR-rG4-bearing mRNAs *(APP, APH1A, HIRA, and LMNB1* mRNAs*)* had significantly longer half-lives under stress (Fig. 4a). In contrast, such a difference was not observed for control RNAs (β-actin and rRNA 18 S) (Supplementary

Fig. 7). Furthermore, we asked whether rG4 stabilizing ligands can confer a similar trend in mRNA stability. Indeed, cPDS and BRACO-19 treated cells showed a similar effect on mRNA stability when compared to the starved cells (Fig. 4b). Collectively, these results suggest that 3'UTR rG4 folding under stress contributes to the stability of corresponding RNAs.

### Reversibility of stress-induced rG4 folding

Reversibility is one of the key features of cellular stress response mechanisms[41]. The inability to reverse stress-induced changes upon stress resolution can perturb cellular physiology and contribute to disease. When folded in vitro, many rG4s represent unusually stable structures that are resistant to elevated temperatures[38], which begs the question of whether in vivo folding of rG4s can be reversed when cells are no longer under stress. To answer this question, we performed a rescue experiment in which the starvation media was replaced with full media at different time points and relative rG4 folding was analyzed. We observed that the elevated cytosolic rG4 signals decreased gradually, and eventually returned to a basal level within an hour after stress removal (Fig. 4c, d, Supplementary Fig. 9), indicating that the in vivo folding of rG4 is dynamic and reversibly regulated by cellular stress. To evaluate whether the unfolding dynamics of stress-induced rG4s upon removal of stress could be modulated by rG4-specific ligands, we treated the recovering cells with cPDS, which delayed the unfolding of cytoplasmic rG4s in recovering cells (Fig. 4c, d). Furthermore, the depletion of cellular ATP levels by inhibition of glycolysis (2-DG) or oxidative phosphorylation (CCCP) also impeded rG4 unfolding upon starvation removal (Fig. 4c, d), suggesting that the dynamic unfolding of stress-induced rG4s is an ATP-dependent process.

## Discussion

While the formation of rG4s in vitro is a well-studied phenomenon, the regulation and function of rG4 folding in living eukaryotic cells are not well understood. Depending on the methods used, previous studies have come to apparently discrepant conclusions regarding rG4 folding in cells. On one hand, studies based on rG4-specific antibodies and probes have shown positive, cytoplasmic, and often RNase-sensitive signals, suggesting that some rG4s are present in cells[9,25]. On the other hand, DMS or SHAPE probing experiments in mammalian and yeast cells have shown that while rG4 structures are readily detected in poly(A)$^+$ transcriptome in vitro, the folding of the same RNAs largely resemble their in vitro rG4-non-permissive (e.g., folded in Li$^+$) state[28]. One possibility that has been previously discussed[29] is that DMS probing would irreversibly trap an rG4 region in an unfolded state, leading to all rG4 regions eventually being unfolded. However, the duration of in vitro and in vivo DMS treatment is usually kept short enough such that even at a stably unfolded state (e.g. Li$^+$), rG4 regions would not be saturated with modifications. In addition, DMS-seq can detect folded rG4s in bacteria[28] and now in human cells under stress, further arguing against the possibility of DMS saturation. Furthermore, unlike DMS, SHAPE preferentially modifies the folded rG4 and would not artificially unfold them, yet still yields similar results to those from DMS probing. To reconcile the apparent discrepancy, we believe that most of the mammalian cell studies are consistent with a revised model in which a vast number of potential rG4s undergo fast folding-unfolding dynamics, with only a small fraction of each rG4 region being in a stably folded state at a given time point in mammalian cells under basal conditions.

Our current study further extends this model and suggests the steady state of rG4 folding/unfolding can be shifted by cellular stress. Interestingly, various stresses tested, including oxidative stress, cold shock, and starvation, could promote the assembly of rG4s, suggesting that different stress sensors may converge on the same or overlapping pathways regulating rG4 folding dynamics in cells. By using both G4 antibodies and G4-specific chemical probes, we demonstrated a stress-dependent shift in the overall rG4 landscape with an increased level of rG4 formation under a variety of stress conditions (Fig. 1 and Supplementary Fig. 1–2). Consistent with these rG4 staining results, our transcriptome-wide DMS-seq analysis identified a large number of folded rG4s in cells under starvation conditions. Since diverse RNA structural elements play a significant role in

RNA stability[42], protein recognition[43], and RNA-protein granule assembly (e.g., stress granules and P-bodies)[17,44,45], we reasoned that stress-induced rG4 folding may also play direct functional roles in mRNA metabolism in human cells. Indeed, we demonstrated that the folding of stress-responsive rG4 elements in 3′UTRs under stress promoted the stability of corresponding mRNAs (Fig. 4). Specifically, we observed that selected rG4-bearing mRNAs showed significantly longer half-lives in stressed cells compared to the control non-rG4-bearing transcripts, suggesting a direct contribution of 3′UTR rG4s to mRNA stability during stress.

Besides the regulation of decay of specific mRNAs, we hypothesize that global rG4 folding may also contribute to other aspects of RNA metabolism in the cells. We propose that rG4s contribute to the preservation of rG4-bearing transcripts during stress, and intermolecular interactions between rG4-bearing transcripts and RNA-binding proteins could also promote the formation of stress-induced RNA-containing biomolecular condensates such as stress granules and P-bodies. Contribution of G4 structures into assembly of biomolecular condensates in vitro and in transfected cells has been reported before[17,46]. Specific subcellular re-localization of rG4-containing transcripts may facilitate the assembly of RNA granules, and upon stress relief such transcript could be readily available for the subsequent mRNA translation.

Although cells respond to environmental stress in diverse ways, these stress-induced changes are reversible upon stress relief and/or adaptation. In fact, the inability to revert gene expression to homeostasis upon resolution of stress contributes to disease[47]. Consistent with this central tenet, we showed that stress-induced rG4 assembly is reversible upon stress removal, implicating rG4 folding as a component of an adaptive stress response program. Our results also show that rG4 unfolding upon stress resolution depends on the cellular energy status suggesting that rG4 disassembly is an active rather than passive process that relies on the action of ATP-dependent machinery, likely RNA helicases such as DHX36[48]. Based on our data, we propose a model (Fig. 4e) that the reversible stress-mediated increase in rG4s folding contributes to mRNA stability. In summary, our study demonstrated the dynamic regulation of rG4 folding in the 3′UTRs of mRNAs in cells and its role as a positive regulator of mRNA stability under cellular stress.

## Methods

### Cell culture, cellular stress, drug treatment, and in vivo DMS treatment

Human osteosarcoma U2OS cells (ATCC, HTB-96) were maintained at 37 °C in a $CO_2$ incubator in DMEM (Corning) supplemented with 10% FBS, 20 mM HEPES (Gibco), 1% penicillin/streptomycin. To induce oxidative stress, cells were treated with 100 uM sodium arsenite (Ars-) in regular media for an hour. Starvation stress was induced by incubating the cells in HBSS media with Ca$^{++}$ and Mg$^{++}$ for 2 h. Dimethyl-sulfate (Sigma) was mixed with ethanol (1:1) and was added to U2OS cells cultured in 15 cm dishes to a final DMS concentration of 8%, and evenly distributed by slow swirling. The cells were incubated for 5 min, the media and excess DMS were decanted, and cells were washed twice with 10 mL of 25% β-mercaptoethanol (Sigma-Aldrich) in PBS to quench any residual DMS. After washing, cells were lysed in 8 mL TRIzol reagent (Invitrogen) supplemented with 5% β-mercaptoethanol, and lysates were immediately processed for mRNA isolation.

Whenever used the cells were treated with 5 μM cPDS, 5 μM PDS, 2 μM QUMA-1, 10 mM 2-DG, 10 μM CCCP, or 10 μg/mL ActD as indicated in the respective assays.

### In vivo rG4 biotinylation

In vitro rG4 biotinylation was validated using the methods previously described (details in Supplementary files)[33]. For in vivo rG4-mediated biotinylation, U2OS cells were treated with 100 μM hemin for 30 mins followed by 100 μM biotin tyramide treatment for 5 min. Next, the cells

were exposed to 100 μM $H_2O_2$ for a minute and the reaction was quenched. Then the cells were PFA-fixed and imaged. Cy3 conjugated streptavidin antibody was used to detect the level of biotinylation in non-stressed vs stressed cells.

## Immunofluorescence

$1 \times 10^5$ U2OS cells were plated into a 24-well plate seed with coverslips. The following day, cells were treated as indicated in figure legends. Then the cells were fixed with 4% paraformaldehyde for 15 min, permeabilized with 0.5% Triton-X 100 for 10 min, and blocked for 45 min with 5% normal horse serum (NHS; ThermoFisher) diluted in PBS. Primary antibodies were diluted in blocking solution and incubated for 1 h at room temperature or overnight at 4 °C. Anti-G-quadruplex antibody (BG4) was purchased from Absolute Antibodies and used at a 1:500 dilution. The cells were washed thrice with PBS and incubated in 1:1000 diluted anti-His tag antibody. Next, cells were washed three times and then incubated with secondary antibodies (Jackson Laboratories) and Hoechst 33258 (Sigma-Aldrich) for 1 h at room temperature and washed. Coverslips were mounted on glass slides with Vinol and imaged.

## Microscopy

Wide-field fluorescence microscopy was performed using an Eclipse E800 microscope (Nikon) equipped with epifluorescence optics and a digital camera (Spot Pursuit USB) or an AXIO observer A1 (Zeiss) equipped with epifluorescence optics and a digital camera (SPOT Idea 5.0 mp). Image acquisition was done with a 40X or 100X objective. Images were merged using Adobe Photoshop and fluorescence intensity was quantified using ImageJ.

## RNA folding

Poly(A)-selected RNA or in vitro synthesized RNA oligos (Integrated DNA technologies) in 150 mM KCl or LiCl in $T_{10}E_{0.1}$ buffer (10 mM Tris-HCl pH 7.4 and 0.1 mM EDTA) were heated to 95 °C for 5 min and then slowly cooled to room temperature.

## RT-stop profiling and analysis of sequencing reads

DMS-seq and data analysis were performed as previously described[28]. Briefly, control or DMS-treated polyA⁺ mRNAs were fragmented by using a buffered zinc solution (Ambion). mRNA fragments were radiolabeled by using T4 polynucleotide kinase (NEB) and separated by denaturing PAGE. 60-80 nt fragments were extracted and ligated to a pre-adenylated 3′ adapter containing the RT priming sequence. After non-ligated adapters were removed by denaturing PAGE, RT was performed by using an RT primer containing the 5′ adapter sequence and SuperScript III (Invitrogen) with either 150 mM K⁺ or Li⁺ at 42 °C for 10 min. After alkaline hydrolysis of the RNA templates, cDNAs were circularized by using circLigase (Lucigen) at 70 °C for 1 h. Sequencing libraries were generated by PCR and gel-purified. Single-end DNA sequencing was performed on a HiSeq2000 using a standard protocol. Sequencing reads were first de-duplicated using SAMtools. After the molecular barcodes and adapter sequences were trimmed, reads were mapped to a hg19 transcriptome reference by Bowtie, requiring zero mismatch and unique mapping. The nucleotide immediately downstream of the most 3′ cDNA nucleotide was assigned as the RT stop. For each transcript, RT stop counts were normalized by the average RT stop count across its entire sequence, generating an RT-stop enrichment score ($E$) for each nucleotide. Nucleotides with $E \geq 10$ (i.e., ≥10-fold higher than average RT stop counts) were assigned as strong RT stops. In vivo folding scores were calculated as $(E_{\text{in vivo}} - E_{\text{in vitro, Li+}})/(E_{\text{in vitro, K+}} - E_{\text{in vitro, Li+}})$.

## Electrophoretic mobility gel shift assay

RNA oligonucleotides were purchased from Integrated DNA Technologies (IDT). To equilibrate, RNAs were diluted to 10 μM in the indicated salt solution, heated to 95 °C for 5 min, and slowly allowed to cool to room temperature. For analysis on acrylamide gels, 10 pmol or 50 pmol were run through a gel and stained with SYBR gold or NMM derivatives, respectively. For SYBR gold staining, gels were post-stained in a solution of 1X SYBR Gold (ThermoFisher Scientific) in 0.5X TBE for 10 min. For NMM staining, gels were post-stained in 100 μM solution of NMM in 0.5X TBE for 30 min. Following staining, gels were destained for 20 min in 0.5X TBE while rocking at room temperature. Gels were visualized using a 265 nm UV transilluminator.

## NMM fluorescence assay

Pre-folded RNA oligos were mixed with 5 μM NMM (or NMM-derivatives) and their fluorescence was measured at 600 nm.

## bNMM pull-down protocol

The bNMM G4 pull-down experiments were performed in 750 μL volume as follows: 100 μM bNMM (5% DMSO) was mixed with 50 μL pre-washed streptavidin agarose beads and tumbled at 4 °C for 15 min. The mixture was centrifuged at 1000 rpm for 1 min and the supernatant was discarded. The beads were washed and suspended in G4-pull-down buffer. Next pre-folded RNA oligos (of pre-folded cellular mRNA) were mixed with bNMM-beads. The mixture was tumbled at 4 °C for 2 h. Next, the bead-RNA mixture was loaded into Biorad columns, and washed with 5*1 mL of G4 buffer (5 min incubation between each wash), before a final wash with G4 buffer at 50 °C. Next, NMM-bound RNA was washed quickly with water, extracted with TRIZOL reagent, and quantified spectrophotometrically.

For the competition experiments, bNMM-rG4 complexes were mixed with 2.5 μM or 5 μM ligands for 1 h at 4 °C before further 1X wash and elution. The eluted oligos were spectrophotometrically quantified and analyzed using polyacrylamide gel electrophoresis.

## cDNA synthesis, qPCR, and PCR

mRNA obtained from bNMM pull-down experiments was reverse transcribed with the Superscript IV first-strand synthesis kit for RT-qPCR (ThermoFisher). The qRT-PCR was performed by using iQ™ SYBR® Green Supermix (Bio-Rad), cDNA template, and gene-specific primer sets designed using the IDT primer design tool. No reverse transcriptase and no template controls were performed in parallel to check for DNA contamination and primer-dimer. The primer sets used for the study are given in Supplementary Table 2. Threshold cycle (CT) values in qRT-PCR experiments were averaged across three biological replicates. The averaged CT value was used to estimate the enrichment of transcripts in the pull-down experiments as well as to estimate their stability in ActD treatment experiments.

## Actinomycin-D treatment and mRNA decay analysis

U2OS cells under different environments (non-stressed, starved, and treated with 10 μM G4 drugs for 2 h) were treated with 10 μg/mL actinomycin-D in growth media for different time points. Total RNA was isolated from the cells. Relative RNA abundance was calculated based on the ratio of CT values in control vs starved cells over time.

## Starvation rescue experiment

U2OS cells under starvation (as described above) were fed with a normal culture medium for different time points and were analyzed for the change in BG4 fluorescence intensity over time.

## Statistics and reproducibility

Statistical parameters are described along with the data. If not stated otherwise, results are mean values ± standard deviation (SD) of at least three independent experiments, or results show one representative experiment of a minimum of three biological replicates. Statistical analyses were performed on all available data. Statistical tests were performed using GraphPad Prism software.

## Drawing tools

Biorender and ChemDraw were used to draw some of the figures.

## Reporting summary

Further information on research design is available in the Nature Portfolio Reporting Summary linked to this article.

## Data availability

The data supporting the findings of this study are available, further information is also available from the corresponding authors upon request. DMS-seq data are deposited in Gene Expression Omnibus (GEO) under the accession GSE200706. Source data are provided with this paper.

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

## Acknowledgements
We thank members of Ivanov, Anderson, and Guo labs for the valuable discussion. National Institutes of Health grant R35 GM126901 (P.J.A.). National Institutes of Health grant R01 GM126150 and R01 GM146997 (P.I.). National Institutes of Health grant DP2 GM132930 (J.U.G.). New York Stem Cell Foundation, NYCSF – Robertson Investigator (J.U.G.).

## Author contributions
Conceptualization: P.I., P.J.A., J.U.G., P.K. Methodology: P.K., M.F., E.V.M., A.V.K., J.U.G. Investigation: P.K., M.F., E.V.M., A.V.K, J.U.G. Funding acquisition: P.I., P.J.A., J.U.G. Project administration: P.I., J.U.G. Supervision: P.I., P.J.A., J.U.G., A.V.K. Writing—original draft: P.I., P.K., A.V.K. Writing—review and editing: P.I., P.J.A., J.U.G., P.K.

## Competing interests
P.J.A. serves on the Scientific Advisory Boards of Simcere Pharmaceutical and Sedec Therapeutics. All other authors declare no competing interests.
