## [Peer Review File · Nature Communications]

Stress promotes RNA G-quadruplex folding in human cellsREVIEWER COMMENTS

Reviewer #1 (Remarks to the Author):

In this manuscript Kharel et al investigate the relationship between rG4 formation and stability with cellular stress. The authors employ a series of strategy, including BG4 IF, DMS-Seq and small-molecule pull-down to measure relative abundance and stability of rG4 in normal and stressed cells. The concept is novel and potentially interesting, but the data reported are not sufficient to support the publication of this manuscript in the current form. If the authors can address the major technical issues of the current study, I believe that this investigation would have the potential to be published in Nat Comm.

Here are my major concerns that should be addressed prior publication:

1) The authors use BG4 total fluorescence in the cytoplasm to assess relative rG4 abundance and stability. The seminal paper cited (Biffi et al 2013 Nat Chem) uses BG4 to probe rG4 counting foci rather than total fluorescence. Total fluorescence cannot be used to assess rG4 prevalence, especially considering what mentioned by the authors on the cross reactivity of BG4 with other structures. This should be validated with a counter staining with small-molecules probes to assess true rG4 prevalence or BG4 foci should be used as in the seminal paper, otherwise this can be highly misleading and cannot be considered conclusive.

2) The conclusion on mRNA stability associated rG4 formation can only be supported if positive and negative controls are provided. Currently the authors are showing only 2 mRNA bearing a G4 and one without, this cannot be used to make general conclusions. This analysis should be extended to more mRNAs otherwise this can simply be due to chance.

3) On the same note of point 2, a key control to add to conclude that is the increased stability of the mRNA is provided by the rG4 will be to perform the same analysis not under stress but upon treatment with G4-ligands that will provide G4 stability in the absence of stress. Without this key control the additional stability observed might be due to different mechanism stress related and not associated to the G4 formation, especially if we are looking at only 3 mRNA transcripts.

4) Figure 3 probably provides the most convincing data, but can recovery mediated by the biotin-NMM be abrogated by addition of a competitive G4 ligand? This is an essential control to validate that the enrichment observed is actually G4-mediated.

5) DMS-treatment is highly dependent on the G4-dynamics and time of incubation with the chemical as demonstrated by the Balasubramanian group in Di Antonio et al Nat Chem 2020, so the authors cannot claim that there are not G4-folded in the absence of stress but rather an increased lifetime of the G4-structure under stress (i.e. not an on/off mechanism).

6) The introduction is scant of details and seminal papers. For example seminal papers for the synthesis

of PDS and cPDS that are used in the manuscript are lacking. Key papers from the Vilar group on rG4 imaging are surprisingly not reported. Furthermore, the authors incorrectly claim that the use of probes such as BG4 or small-molecules ligands limits the study of G4 as these probes can perturb the in cells folding of G4 structures. This has been address by using single-molecule imaging (Di Antonio et al Nat Chem 2020), so the authors here are providing a partial report of the literature that is misleading and inaccurate.

7) The manuscript is far too short, not enough details are provided in the results section and the discussion is kept to the bare minimum. This makes the manuscript hard to follow, especially considering the amount of data actually provided. The authors should make a better effort to articulate their findings and guide the reader through the data.

Providing that the authors can address these concern in full, I believe that a revised version of this manuscript could have the potential to be published in Nat Comm

Reviewer #2 (Remarks to the Author):

The regulation and function of RNA quadruplex (rG4) folding in living human cells is not well understood. In this manuscript, Ivanov and colleagues demonstrate that that the folding of potential RNA quadruplexes in human cells into real rG4 structures is dynamically regulated by stress. In addition, the authors suggest that rG4 found in the 3'UTR regulate mRNA stability during stress.

Overall, this is an interesting and novel report on the role of RNA quadruplexes on RNA fate. I recommend acceptance pending minor revision:

“rG4s become more folded upon starvation” Is this related to autophagy induction? Several reports (PMID: 19996277, PMID: 22627293, PMID: 30759257) have found that G4 ligands induce autophagy. What happens in autophagy-deficient cells?

Figure 1a: this is a detail, but the representation of the parallel quadruplex is incorrect: due to G4 right-handed helicity the chain reversal loop should be “N” shaped, while a “mirror N” is shown here.

Figure 1b-1d: perhaps add some close-up of individual cells (as sup info ?)

Figure 2b: define what a “strong RT stop” is as compared to a regular one.

Figure 2d: define how the boundary between black and grey points was defined

Figure 2ef: “in vivo rG4 folding increases significantly in starved U2OS cells but not in the unstressed cells” May be rephrase: “in vivo rG4 folding increases significantly in starved U2OS cells as compared to the unstressed cells”?

Figure 3c: would be nice to present a non G4 forming RNA for comparison

Figure 3g: not sure why Ct values are mentioned in the legend: they are not used for this figure. In

addition, I am surprised that the level of significance (*) (whatever that means) is the same for HIRA and CDKN1B?

Figure 3h : gel is spliced – mark the boundary clearly – in addition legend above the gel does not exactly match the position of the bands; please check.

“elevated cytosolic rG4 signals were also decreased gradually, and eventually returned to a basal level within an hour of stress removal (indicating that the in vivo folding of rG4 is dynamic and reversibly regulated by cellular stress” Is this because the G4 within a specific RNA is opened, or because these RNA are replaced by newly-synthesized ones in which rG4 was not formed?.

“their folding can be rapidly induced by stress” Define “rapid” or provide a time scale. Within minutes? Hours?

All results were obtained in U2OS cells. Any evidence that the same results would be obtained in other cell types?

Typo: (page 1) forrG4

We thank the reviewers for their thoughtful comments. Our response to the comments are included in red.

Reviewer #1 (Remarks to the Author):

In this manuscript Kharel et al investigate the relationship between rG4 formation and stability with cellular stress. The authors employ a series of strategy, including BG4 IF, DMS-Seq and small-molecule pull-down to measure relative abundance and stability of rG4 in normal and stressed cells. The concept is novel and potentially interesting, but the data reported are not sufficient to support the publication of this manuscript in the current form. If the authors can address the major technical issues of the current study, I believe that this investigation would have the potential to be published in Nat Comm.

Here are my major concerns that should be addressed prior publication:

1) The authors use BG4 total fluorescence in the cytoplasm to assess relative rG4 abundance and stability. The seminal paper cited (Biffi et al 2013 Nat Chem) uses BG4 to probe rG4 counting foci rather than total fluorescence. Total fluorescence cannot be used to assess rG4 prevalence, especially considering what mentioned by the authors on the cross reactivity of BG4 with other structures. This should be validated with a counter staining with small-molecules probes to assess true rG4 prevalence or BG4 foci should be used as in the seminal paper, otherwise this can be highly misleading and cannot be considered conclusive.

Response: The reviewer correctly pointed out the possibility of overestimating rG4 folding by BG4 quantitation alone. Regarding the quantification of BG4 foci vs overall intensity, it is unclear to us why the rG4s within soluble cytoplasmic mRNAs would appear as individual foci unless they aggregate, for example, in stress granules. In our experiments (with the cell types and fixing method we used), we did not observe foci staining. To confirm that the observed increase in BG4 signal was indeed due to rG4s, we have taken two additional complementary approaches: 1) rG4 staining by a chemical dye QUMA-1 (Fig. 1e) and 2) a biotinylation assay based on the peroxidase-like activity of hemin-bound G4s to biotinylate neighboring biomolecules both *in vitro* and *in vivo* (Fig. 1f-g). Both approaches confirmed our BG4 findings, thus further supporting BG4 staining as a fair representation of the cellular rG4 folding landscape in our experiments.

2) The conclusion on mRNA stability associated rG4 formation can only be supported if positive and negative controls are provided. Currently the authors are showing only 2 mRNA bearing a G4 and one

without, this cannot be used to make general conclusions. This analysis should be extended to more mRNAs otherwise this can simply be due to chance.

Response: We have now expanded our analysis to 4 rG4 mRNA candidates, and observed similar results (Fig.4a).

3) On the same note of point 2, a key control to add to conclude that the increased stability of the mRNA is provided by the rG4 will be to perform the same analysis not under stress but upon treatment with G4-ligands that will provide G4 stability in the absence of stress. Without this key control the additional stability observed might be due to different mechanism stress related and not associated to the G4 formation, especially if we are looking at only 3 mRNA transcripts.

Response: This is an excellent suggestion. We have now incorporated mRNA stability analysis after the treatment with G4 ligands, namely cPDS and BRACO19. Indeed, rG4 ligands could stabilize the rG4-containing mRNAs without stress (Fig. 4b).

4) Figure 3 probably provides the most convincing data, but can recovery mediated by the biotin-NMM be abrogated by addition of a competitive G4 ligand? This is an essential control to validate that the enrichment observed is actually G4-mediated.

Response: Following this suggestion, we have done competition experiments with several G4 ligands (Fig. 3f). We observed that two of the tested ligands (BRACO-19, and PhenDC3) can compete off rG4s from the NMM bound complexes while the other two ligands (PDS and cPDS) did not compete off the bound rG4s from rG4-bNMM complexes. The latter can be explained based on the following two facts- (1) the dissociation constant (K_d) of G4-PDS (K_d : ~450 nM, Mandal et al, <https://doi.org/10.1093/nar/gkz135>) is reportedly larger than G4-NMM (K_d : ~100 nM, Perenon et al, <https://doi-org.ezp-prod1.hul.harvard.edu/10.1039/C9CP06321H>) binding, and (2) They might interact with the G4s via a different mechanism.

5) DMS-treatment is highly dependent on the G4-dynamics and time of incubation with the chemical as demonstrated by the Balasubramanian group in Di Antonio et al Nat Chem 2020, so the authors cannot claim that there are not G4-folded in the absence of stress but rather an increased lifetime of the G4-structure under stress (i.e. not an on/off mechanism).

Response: Great point. We have now modified the text accordingly.

6) The introduction is scant of details and seminal papers. For example seminal papers for the synthesis of PDS and cPDS that are used in the manuscript are lacking. Key papers from the Vilar group on rG4 imaging are surprisingly not reported. Furthermore, the authors incorrectly claim that the use of probes such as BG4 or small-molecules ligands limits the study of G4 as these probes can perturb the in cells folding of G4 structures. This has been address by using single-molecule imaging (Di Antonio et al Nat Chem 2020), so the authors here are providing a partial report of the literature that is misleading and inaccurate.

Response: We have modified the introduction and added essential references

7) The manuscript is far too short, not enough details are provided in the results section and the discussion is kept to the bare minimum. This makes the manuscript hard to follow, especially considering the amount of data actually provided. The authors should make a better effort to articulate their findings and guide the reader through the data.

Response: We appreciate the suggestion and have revised both the Results and Discussion in the revised manuscript.

Providing that the authors can address these concern in full, I believe that a revised version of this manuscript could have the potential to be published in Nat Comm

Thank you!

Reviewer #2 (Remarks to the Author):

The regulation and function of RNA quadruplex (rG4) folding in living human cells is not well understood. In this manuscript, Ivanov and colleagues demonstrate that that the folding of potential RNA quadruplexes in human cells into real rG4 structures is dynamically regulated by stress. In addition, the authors suggest that rG4 found in the 3'UTR regulate mRNA stability during stress.

Overall, this is an interesting and novel report on the role of RNA quadruplexes on RNA fate. I recommend acceptance pending minor revision:

Response: We thank the reviewer for the positive evaluation of our work.

“rG4s become more folded upon starvation” Is this related to autophagy induction? Several reports (PMID: 19996277, PMID: 22627293, PMID: 30759257) have found that G4 ligands induce autophagy. What happens in autophagy-deficient cells?

Response: We think that there is a great possibility that autophagy is involved. Unfortunately, we have no access to autophagy-deficient cells, and this is not our expertise in general. However, we will try to do some pilot experiments on this point in future.

Figure 1a: this is a detail, but the representation of the parallel quadruplex is incorrect: due to G4 right-handed helicity the chain reversal loop should be “N” shaped, while a “mirror N” is shown here.

Figure 1b-1d: perhaps add some close-up of individual cells (as sup info ?)

Response: Fixed.

Figure 2b: define what a “strong RT stop” is as compared to a regular one.

Figure 2d: define how the boundary between black and grey points was defined

Response: This information is now added to “Materials and Methods” and figure legends.

Figure 2ef: “in vivo rG4 folding increases significantly in starved U2OS cells but not in the unstressed cells” May be rephrase: “in vivo rG4 folding increases significantly in starved U2OS cells as compared to the unstressed cells”?

Response: Done.

Figure 3c: would be nice to present a non G4 forming RNA for comparison

Response: Done.

Figure 3g: not sure why Ct values are mentioned in the legend: they are not used for this figure. In addition, I am surprised that the level of significance (*) (whatever that means) is the same for HIRA and CDKN1B?

Response: Fixed now.

Figure 3h : gel is spliced – mark the boundary clearly – in addition legend above the gel does not exactly match the position of the bands; please check.

Response: Corrected now.

“elevated cytosolic rG4 signals were also decreased gradually, and eventually returned to a basal level within an hour of stress removal (indicating that the in vivo folding of rG4 is dynamic and reversibly regulated by cellular stress” Is this because the G4 within a specific RNA is opened, or because these RNA are replaced by newly-synthesized ones in which rG4 was not formed?.

Response: Both scenarios are theoretically possible but it is likely to be a mixed scenario. It is clear that the bulk of mRNAs is already present in the cytosol. Only limited fraction of newly synthesized mRNAs is exported to the cytoplasm within the experimental window of 1 hour. Additionally, another important aspect to consider is an average half-life of an average mRNA in mammalian cells, which is ~10 hours (with fast-decaying fraction of mRNAs of ~2hr half-life (PMID: **12902380**)). Thus it is unlikely that the pre-existing cytosolic mRNAs are degraded and substituted by newly exported within 1 h.

“their folding can be rapidly induced by stress” Define “rapid” or provide a time scale. Within minutes? Hours?

Response: Thank you for this point. For post-transcriptional responses such as stress granule disassembly minutes are enough (10-60 mins). However, for the purpose of the clarity we have now removed “rapid” here

All results were obtained in U2OS cells. Any evidence that the same results would be obtained in other cell types?

Response: We observed similar results in COS7 cells. Included as supplementary material.

Typo: (page 1) forrG4

Response: Fixed

REVIEWERS' COMMENTS

Reviewer #1 (Remarks to the Author):

The authors have successfully addressed the key criticisms I have raised and I am therefore supportive of publication of the revised manuscript in Nat Commun.

Reviewer #2 (Remarks to the Author):

The authors have addressed my comments in a satisfactory manner. I recommend acceptance.

(note, however, that the schematic representation of a parallel quadruplex is correct in Figure 4e, but not in figures 1a-1f-2a-3g... mirror image.